# Salvage Radical Prostatectomy for Recurrent Prostate Cancer: A Systematic Review (French ccAFU)

**DOI:** 10.3390/cancers15225485

**Published:** 2023-11-20

**Authors:** Amine Saouli, Alain Ruffion, Charles Dariane, Eric Barret, Gaëlle Fiard, Gaelle Fromont Hankard, Gilles Créhange, Guilhem Roubaud, Jean Baptiste Beauval, Laurent Brureau, Raphaëlle Renard-Penna, Mathieu Gauthé, Michael Baboudjian, Guillaume Ploussard, Morgan Rouprêt

**Affiliations:** 1Department of Urology, CHU Souss Massa, Faculty of Medicine and Pharmacy, Ibn Zohr University, Agadir 80000, Morocco; 2Service D’urologie Centre Hospitalier Lyon Sud, Hospices Civils de Lyon, 69002 Lyon, France; alain.ruffion@chu-lyon.fr; 3Équipe 2, Centre D’innovation en Cancérologie de Lyon (EA 3738 CICLY), Faculté de Médecine Lyon Sud, Université Lyon 1, 69002 Lyon, France; 4Comité de Cancérologie de l’Association Française d’Urologie, Groupe Prostate, Maison de l’Urologie, 11, Rue Viète, 75017 Paris, France; gaellef@gmail.com (G.F.); mathieugauthe@yahoo.fr (M.G.); 5Department of Urology, Hôpital Européen Georges-Pompidou, AP-HP, Paris University, U1151 Inserm-INEM, F-75015 Paris, France; dcharlie8@hotmail.com; 6Department of Urology, Institut Mutualiste Montsouris, 42 Boulevard Jourdan, 75014 Paris, France; eric.barret@imm.fr (E.B.); laurent.brureau@chu-guadeloupe.fr (L.B.); 7Department of Urology, Grenoble Alpes University Hospital, Université Grenoble Alpes, CNRS, Grenoble INP, TIMC-IMAG, 38400 Grenoble, France; 8Department of Pathology, CHRU Tours, 37000 Tours, France; gaelle.fromont-hankard@univ-tours.fr; 9Department of Radiotherapy, Institut Curie, 75005 Paris, France; gilles.crehange@curie.fr; 10Department of Medical Oncology, Institut Bergonié, 33000 Bordeaux, France; g.roubaud@bordeaux.unicancer.fr; 11Department of Urology, CHU Rennes, CIC 1414 Rennes, France; jbbeauval@gmail.com; 12AP-HP, Radiology, Pitie-Salpetriere Hospital, Sorbonne University, 75013 Paris, France; raphaele.renardpenna@aphp.fr; 13Service D’urologie et de Transplantation Rénale, CHU La Conception, 13005 Marseille, France; michael.baboudjian@outlook.fr; 14Department of Urology, La Croix du Sud Hôpital, 31130 Quint-Fonsegrives, France; g.ploussard@gmail.com; 15GRC 5 Predictive Onco-Uro, AP-HP, Urology, Pitie-Salpetriere Hospital, Sorbonne University, 75013 Paris, France; mroupret@gmail.com

**Keywords:** salvage radical prostatectomy, recurrence, prostate cancer, systematic review

## Abstract

**Simple Summary:**

We reviewed the available studies assessing salvage surgery after recurrent prostate cancer with primary non-surgical treatment. While the studies used had the potential for bias, due to their retrospective type, we looked at treatment outcomes and toxicity for men treated with a number of salvage radical prostatectomies for recurrent prostate cancer. We demonstrated that SRP can be considered a suitable treatment option for selected patients.

**Abstract:**

The aim of this study was to systematically review the current evidence regarding the oncological and functional outcomes of salvage radical prostatectomy (sRP) for recurrent prostate cancer. A systematic review was conducted throughout September 2022 using the PubMed, Science Direct, Scopus, and Embase databases. Preferred Reporting Items for Systematic Reviews and Meta-analysis (PRISMA) guidelines were followed to identify eligible studies. A total of 55 studies (3836 patients) met our eligibility criteria. The vast majority of men included had radiation therapy (including brachytherapy) as their first-line treatment (*n* = 3240, 84%). Other first-line treatments included HIFU (*n* = 338, 9%), electroporation (*n* = 59, 2%), proton beam therapy (*n* = 54, 1.5%), cryotherapy (*n* = 34, 1%), focal vascular targeted photodynamic therapy (*n* = 22, 0.6%), and transurethral ultrasound ablation (*n* = 19, 0.5%). Median preoperative PSA, at the time of recurrence, ranged from 1.5 to 14.4 ng/mL. The surgical approach was open in 2300 (60%) cases, robotic in 1465 (38%) cases, and laparoscopic in 71 (2%) cases. Since 2019, there has been a clear increase in robotic versus conventional surgery (1245 versus 525 cases, respectively). The median operative time and blood loss ranged from 80 to 297 min and 75 to 914 mL, respectively. Concomitant lymph node dissection was performed in 2587 cases (79%). The overall complication rate was 34%, with a majority of Clavien grade I or II complications. Clavien ≥ 3 complications ranged from 0 to 64%. Positive surgical margins were noted in 792 cases (32%). The median follow-up ranged from 4.6 to 94 months. Biochemical recurrence after sRP ranged from 8% to 51.5% at 12 months, from 0% to 66% at 22 months, and from 48% to 59% at 60 months. The specific and overall survival rates ranged from 13.4 to 98% and 62 to 100% at 5 years, respectively. Urinary continence was maintained in 52.1% of cases. sRP demonstrated acceptable oncological outcomes. These results, after sRP, are influenced by several factors, and above all by pre-treatment assessment, including imaging, with the development of mpMRI and metabolic imaging. Our results demonstrated that SRP can be considered a suitable treatment option for selected patients, but the level of evidence remains low.

## 1. Introduction

Prostate cancer (PCa) is the most commonly diagnosed cancer in men, with an estimated 1.4 million diagnoses recorded worldwide in 2020 [1]. Although active surveillance is increasingly used, most PCa patients undergo definitive local treatment, followed by prostate-specific antigen (PSA) monitoring [2]. However, it is estimated that 27% to 53% of all patients undergoing radical prostatectomy (RP) or radiation therapy (RT) develop biochemical recurrence (BCR) [3]. While there is a standard treatment pathway for post-RP BCR, there is no widely adopted treatment paradigm for BCR after primary nonsurgical treatment. In addition, there have been no randomized trials comparing the oncological outcomes of available salvage therapies, and thus, there is no clear consensus regarding the best treatment option. As such, many patients with BCR after primary nonsurgical treatment receive androgen deprivation therapy (ADT), which denies them any chance of curative therapy [4].

Salvage radical prostatectomy (sRP) is a challenging procedure that is rarely performed, although it represents a guidelines-validated option for BCR after primary nonsurgical treatment. The historical series of sRP with frequent major complications, such as rectal injury and poor functional outcomes [5], have played a major role in the low use of this option in a salvage situation. However, minimally invasive approaches may provide significant improvements, which could lead to improved functional outcomes and reduced complications [6,7]. With the renewed interest in sRP, identifying patients who would benefit most from sRP is crucial to avoid overtreatment and limit treatment-related toxicities.

In this study, we aimed to systematically review the current evidence regarding the oncological and functional outcomes of sRP for recurrent PCa after primary nonsurgical treatment.

## 2. Methods

### 2.1. Search Strategy

We conducted a systematic review in line with the Preferred Reporting Items for Systematic Reviews and Meta-analyses (PRISMA) guidelines [8].

This protocol was registered in the International Prospective Register of Systematic Reviews (PROSPERO) database (Registration Number: CRD42022378227). We conducted a literature search in PubMed/Medline, Embase, and Science Direct databases, to identify reports published through September 2022, which addressed the oncological and functional outcomes of sRP. The search strategy included the following MeSH terms: *prostatectomy*, *Prostate Cancer*, *Neoplasm Recurrences*, *treatment, Local, Radiation Therapy, Cryotherapies, Salvage Treatment*, *Robot-Assisted Surgery*, and *Surgical Procedure*. Initial screening was independently performed by two investigators (A.S. and G.P.) based on the titles and abstracts of the articles to identify ineligible reports. Reasons for exclusions were noted. Potentially relevant reports were subjected to a full-text review, and the relevance of the reports was confirmed after the data extraction process. Disagreements were resolved by consultation with a third co-author (M.B.).

### 2.2. Study Selection

Studies were deemed eligible if they included men with recurrent PCa after primary nonsurgical treatment (patients), managed with sRP (intervention), and if they assessed oncological and/or functional outcomes (outcome) in randomized controlled trials, nonrandomized prospective studies, and retrospective studies (study design). In case of duplicate publications, either the higher-quality or the most recent publication was selected. Reviews, meta-analyses, editorials, commentaries, authors’ replies, meeting abstracts of unpublished studies, and case reports were excluded, but the reference section was checked for relevant articles. No restriction on the publication language was applied. We searched reports published between January 2008 to September 2022 (Supplementary Material Appendix A).

### 2.3. Data Extraction

Data on studies, patients, treatment, and follow-up were independently extracted by two authors (A.S. and G.P.). We extracted the following variables from the included studies: first author’s name, publication year, sample size, age, pre-sRP PSA, pre- and post-sRP TNM stage, International Society of Urological Pathology (ISUP) score at pre-sRP biopsy, surgical approach, operative time, estimated blood loss, rate and severity of postoperative complications according to the Clavien-Dindo classification, rate of urinary continence, follow-up data, BCR rates, cancer-specific survival, and overall survival.

### 2.4. Assessment of Methodological Quality

Two authors (A.S. and G.P.) independently assessed the quality of the studies and the risk of bias. The risk of bias was assessed according to EAU recommendations for performing systematic reviews and meta-analysis [9]. The Quality Appraisal tool for case series using a Modified Delphi technique was used for retrospective studies [10].

## 3. Results

### 3.1. Study Selection and Characteristics

The study selection process is outlined in the PRISMA flow diagram (Figure 1). A total of 403 full-text articles were assessed for eligibility and 55 met our inclusion criteria [7,11,12,13,14,15,16,17,18,19,20,21,22,23,24,25,26,27,28,29,30,31,32,33,34,35,36,37,38,39,40,41,42,43,44,45,46,47,48,49,50,51,52,53,54,55,56,57,58,59,60,61,62,63,64].

The baseline characteristics of the included studies are summarized in Table 1. A total of 3836 patients were included, ranging from 4 to 428 patients per study. The median age of the patients ranged from 59.5 to 71 years, and the median preoperative PSA ranged from 1.5 to 14.4 ng/mL. The vast majority of men included had RT as first-line treatment [Brachytherapy (BT) 632 (16.5%), external-beam radiation therapy (EBRT) 1878 (49%) and BT/EBRT 121 (3%)]. High-Intensity Focused Ultrasound (HIFU) in three hundred and thirty-eight (9%) cases, electroporation in fifty-nine (2%) cases, Proton Beam Therapy (PBT) in fifty-four (1.5%) cases, cryotherapy in thirty-four (1%) cases, focal Vascular Targeted Photodynamic therapy (VTP) in twenty-two (0.6%) cases, transurethral ultrasound ablation (TULSA) in nineteen (0.5%) cases, cryoablation in three (0.07%) cases, cyberknife in two (0.05%) cases, laser ablation in thirteen (0.3%) cases, Cobalt therapy in two (0.05%) cases, and tomography in (0.02%) case. ISUP ≥ 4 was present in 0 to 70% of cases on the initial diagnostic biopsies. Neo-adjuvant androgen deprivation therapy (ADT) at the time of recurrence was used in 266 cases (17%).

### 3.2. Perioperative Results

The perioperative data are presented in Table 2. Regarding the surgical approach, an open approach was used in 2300 cases (60%), a robotic approach in 1465 cases (38%), and a laparoscopic approach in 71 cases (2%); but since 2019, there has been more frequent use of robotic versus conventional surgery (1245 versus 525 of cases, respectively).

A total of 45 studies reported data on concomitant lymph node dissection. The median number of nodes yielded was reported in 14 studies and ranged from 6 to 17, including 593 (20.5%) patients which were staged pN+ at final pathology.

Regarding pathological features, stage ≥pT3, positive surgical margins, and pN+ status ranged from 5 to 75%, 25 to 82%, and 3 to 60%, respectively. The pathological Gleason score was ≥8 in 6 to 67% of cases. These data were missing in nine studies.

### 3.3. Complications and Functional Results

The reported postoperative complications are summarized in Table 2. The overall complication rate was 34%; with a majority of Clavien grade I or II complications. Clavien grade ≥3 complications ranged from 0 to 64%. The complete urinary continence rate (no pad use) was 52.1% (Table 3). The rates of urinary continence were 56% and 47%, respectively, in minimally invasive (i.e., laparoscopic and robotic) and open approaches.

The urinary continence rate in the primary non-radiation-treatment group (HIFU, electroporation, proton beam therapy, cryotherapy, focal vascular targeted photodynamic therapy, and transurethral ultrasound ablation) was 67% versus 55% in patients formerly treated by RT.

### 3.4. Oncological Results

The median follow-up ranged from 4.6 to 94 months (36, 22, and 39.5 months, respectively, in laparoscopic, robotic, and open approaches). Biochemical recurrence ranged from 8% to 51.5% at 12 months, from 0% to 66% at 22 months, and from 48% to 59% at 60 months. Specific and overall survival rates ranged from 13.4 to 98% and 62 to 100% at 5 years, respectively (Table 3).

The rates of BCR were 20%, 27%, and 47%, respectively, in laparoscopic, robotic, and open approaches. Overall survival was 100% and 98% in the laparoscopic and robotic groups, respectively, and 74% in the open surgery group. The rates of BCR were 36% and 21% in the group of patients treated by non-radiation therapy and RT, respectively. Overall survival was 98% in the group of other primary treatments and 85% for patients treated by RT.

## 4. Discussion

sRP for recurrent PCa after primary non-surgical treatment failure is challenging for urologists due to its aggressive features and technical demands. The majority of PCa patients who present recurrent disease after RT are therefore treated with palliative ADT while a salvage treatment initiated early may change the disease course. As a result, only 1% of the patients recurring after RT indeed undergo salvage surgery [65].

In the present systematic review, we found that sRP may represent a good alternative that can be provided to carefully selected patients. It may lead to a durable response if initiated early and may delay progression and use for systemic therapies.

The introduction of minimally invasive approaches regarding sRP could be associated with many advantages, such as decreasing the rates of overall and high-grade complications (i.e., Clavien > 2). The robotic approach has been also associated with lower rates of blood loss, rectal injury, anastomotic stricture, and postoperative incontinence [66]. Recently, it has been suggested that the Retzius-sparing approach could also be interesting as it allows a meticulous dissection near the often fibrotic rectal plane. Using this approach, Madi et al. only noted one intraoperative urine leak in their salvage Retzius-sparing (SRS) group [12]. Taken together, the implementation of a minimally invasive approach in sRP has led to a renewed interest in this option for managing recurrence after the primary nonsurgical management of PCa.

One of the major limitations attributed to sRP is the poor functional outcomes regarding the urinary continence associated with this option. Thus, we found an overall complication rate of 34%, including rectal wounds, ureteral complications, rectourethral fistula, lymphoceles, anastomotic leakage, and urinary tract infections, which is in line with a previous report from Matei et al., who reported a Clavien > 2 complication rate of 0–33% [66].

However, the functional results widely differed between the studies included in this systematic review. Continence rates reported after sRP ranged from 10 to 100%. This heterogeneity could be explained, once again, by the surgical approach used. Robotic-assisted sRP appeared to improve the early return to continence, compared to open surgery series. This is thought to be due to the support of the surrounding ligaments to the anterior urethra, which helps to maintain sphincteric integrity after SRS [67,68]. Mason et al. suggested that continence outcomes were significantly improved in the SRS group for the treatment of radioresistant prostate cancer [69].

Oncological outcomes after sRP are influenced by several factors and may vary depending on the patient/tumor characteristics, type of initial treatment, surgical approach used, length of follow-up, and, above all, pre-treatment assessment (including imaging, with the development of MRI and metabolic imaging). At the mid-term follow-up, we found that the oncological outcomes were acceptable, as a significant proportion of men were disease-free after five years (i.e., the BCR-free survival rate ranged from 48% to 59% at 60 months). In addition, cancer-specific survival and overall survival rates ranged from 13.4 to 98% and 62 to 100% at 5 years, respectively. However, long-term data remain poorly reported in the literature. Two series showed a 10-year BCR-free survival of 31% and 37%, respectively [25,36]. We therefore encourage further studies evaluating long-term oncological outcomes in these patients.

Our study has several strengths, including the important number of studies/patients included, with a variety of nonsurgical primary treatments with a clear distinction between them, the inclusion of most updated data, and their careful review for study inclusion.

Some limitations must be acknowledged. The main limitation is the significant risk of bias, as all included studies were retrospective, which prevented us from reaching a high level of evidence and from providing clear recommendations. Finally, the heterogeneity regarding the surgical approach used, the type of initial local, and the functional erectile results, which are not reported in our review, are important limitations to notice. Of note, although we performed a systematic review, a meta-analysis was not possible given the heterogeneity of the studies in terms of the initial treatment proposed and the surgical approach.

## 5. Conclusions

sRP appears to be feasible with acceptable morbidity in well-selected PCa patients who recur after primary non-operative surgical treatment. The development of a minimally invasive approach and the improvement of surgical techniques are considered to be two key factors in improving perioperative outcomes. However, the level of evidence remains low as comparative and long-term data are lacking.

## Figures and Tables

**Figure 1 cancers-15-05485-f001:**
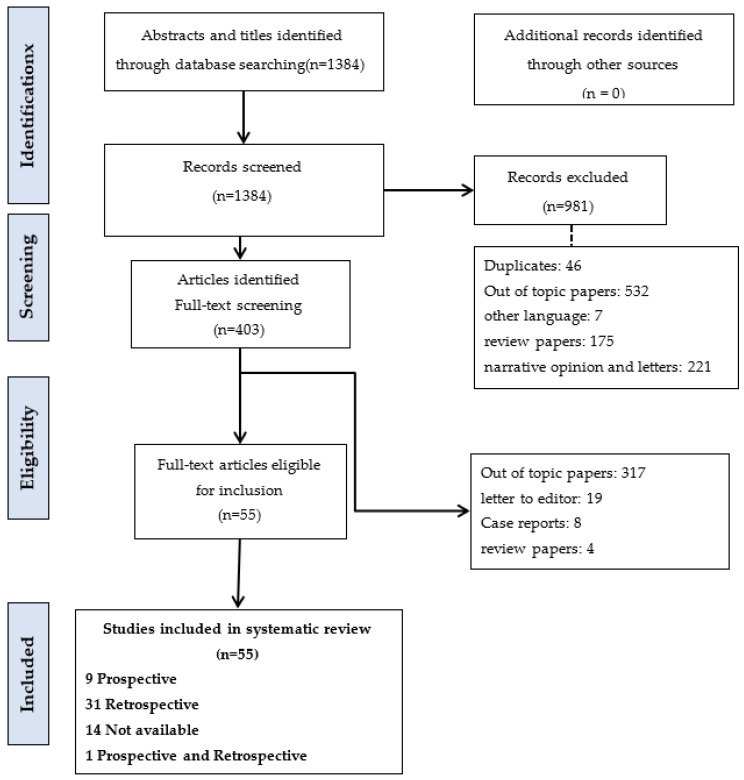
Systematic review PRISMA flow diagram.

**Table 1 cancers-15-05485-t001:** Pre-operative characteristics of the included studies.

Authors (Years)	Number of Patients	Study Type	Inclusion Criteria	Age Median, Years	Initial Local Therapy Type *n* (%)	Pre-sRP ADT *n* (%)	Clinical Staging *n* (%)		Pre-sRP Biopsy *n* (%) ≥ISUP 4	Pre-sRP PSA Median (ng/mL)
							≤T2	T3≥		
Kaouk et al. (2008) [13]	4	Retrospective	-a life-expectancy of >10 years -biopsy confirmed recurrence of PCa	NA	-BT 2 (50) -BT/EBRT 2 (50)	2 (50)	3 (75)	NA	1 (25)	Mean 3.84
Liatsikos et al. (2008) [14]	12	NA	-proven biochemical failure of other alternative therapeutic approaches	Mean 63.3	-HIFU 4 (33) -EBRT 6 (50) -BT 2 (17)	NA	NA		NA	Mean 12.7
Kim et al. (2008) [15]	7	NA	Biopsy proven, local recurrences	Mean 65.5	-EBRT 6 (86) -IMRT 1 (14)	NA	NA	NA	NA	Mean 14.3
Boris et al. (2009) [16]	11	Retrospective	-TRUS-guided prostate biopsies that showed persistent cancer after RT -negative preoperative CT and bone scans	Mean 65	-BT6 (55) -EBRT 3 (27) -BT/EBRT 1 (9) -IMRT 1 (9)	0 (0)	11 (100)	0 (0)	3 (27)	Mean 5.2
Seabra et al. (2009) [17]	42	Prospective	-biopsy confirmed Recurrence of PCa	61	-EBRT	NA	42 (100)	0 (0)	3 (7)	Mean 1.5
Leonardo et al. (2009) [18]	32	NA	a life expectancy of more than 10 years, absence of systemic disease and persistent PCa detected by biopsy	63	-EBRT	5 (16)	25 (78)	7 (22)	12 (37.5)	13
Nunez-Mora et al. (2009) [19]	9	NA	All recurrence was histologically confirmed	59.3	-BT 5 (55.5) -EBRT 4 (44.5)	1 (11)	NA		6 (67)	9.1
Paparel et al. (2009) [20]	146	Retrospective	-A life expectancy >10 yr -clinically localized prostate cancer determined by biopsy -absence of significant voiding symptoms or urinary incontinence -a negative evaluation for systemic disease	65	radiation therapy	NA	58 (42)	79 (58)	16 (11)	5.1
Heidenreich et al. (2010) [21]	55	NA	-A life expectancy >10 yr -clinically organ-confined disease -Absence of locoregional and systemic metastases, -PSA 20 ng/mL		(3D) EBRT: 19 (34.5) EBRT + BT: 15 (27.5) Seed implantation: 21 (38)	NA	44 (80)	11 (20)	10 (18.)	<10: 45 (82) 10.1–20: 10 (18)
Strope et al. (2010) [22]	6	Prospective	-biopsy documented locally recurrent or radiation-resistant prostate cancer -Only patients with no evidence of metastatic disease on bone scan and CT scan	NA	-BT 2 (34) -EBRT 4 (66)	NA	NA		2 (34)	Mean 9.3
Eandi et al. (2010) [23]	18	Retrospective	-Biochemical failure after irradiation	67	-BT 8 (44) -EBRT 8 (44) -PBT 2 (12)	4 (22)	NA		6 (33)	6.8
Chauhan et al. (2011) [24]	15	Retrospective	-had biopsy-proven recurrent PCa	62	-EBRT 5 (33) -BT 5 (33) -PBT 2 (14) -XRT + BT 3 (20)	NA	15 (100)	0 (0)	3 (20)	3.6
Chade et al. (2011) [25]	404	Retrospective	-biopsy confirmed Recurrence of PCa	65	-BT, EBRT 11 (3) -BT, EBRT, IMRT 2 (0) -BT alone 76 (19) -EBRT, 3-DCRT 5 (1) -EBRT, IMRT 5 (1) -EBRT alone 253 (63) -Unknown 52 (13)	0 (0)	262 (55)	72 (18)	80 (20)	4.5
Gorin et al. (2011) [26]	24	Retrospective	life expectancy of at least 10 years and a negative metastatic workup	Mean 64.5	-EBRT 13 (54) -BT 11 (46)	14 (58)	NA		9 (37.5)	Mean 8.7
Ahallal et al. (2011) [27]	15	Retrospective	biopsy-proven local recurrence after cryotherapy or radiation therapy for localized prostate cancer	62.3	-EBRT 8 (53) -BT 6 (40) -cryotherapy 1 (7)	NA	12 (80)	1 (7)	4 (27)	3.5
Lawrentschuk et al. (2011) [28]	15	Prospective	men presenting with an increasing PSA and biopsy-proven PC after primary therapy with HIFU	64	HIFU	1 (7)	15 (100)	0 (0)	0 (0)	7
Leonardo et al. (2012) [29]	13	Retrospective	biopsy-proven local recurrence after HIFU	61.3	HIFU	NA	13 (100)	0 (0)	0 (0)	3.31
Kaffenberger et al. (2013) [30]	34	Retrospective	failure of prior definitive therapy	66.5	-BT 13 (38) -EBRT 11 (32) -combined BT/EBRT 6 (18) -HIFU 4 (12)	4 (12)	32 (94)	2 (6)	12 (35)	3.86
Peters et al. (2013) [31]	44	Retrospective	All men showed PSA failure afterward, and recurrences were confirmed by biopsies	65	-EBRT 31 (70) -BT 2 (5) -I-125 11 (25) -IMRT 0 (0)	5 (11)	30 (69)	14 (31)	7 (16)	0–10 24 (55) >10–20 18 (41) >20 2 (5)
Yuh et al. (2014) [32]	51	Prospective	-BCR -biopsy confirmed recurrence of PCa -negative CT results and bone scans	68	-BT 22 (43.1) -BT+EBRT 1 (2) -Cryoablation 3 (5.9) -EBRT 18 (35.3) -HIFU 1 (2.0) -PBT 6 (11.8)	18 (19.6)	NA		NA	5.27
Zugor et al. (2014) [33]	13	Retrospective	-radiation-resistant PCa	63	-EBRT 7 (54) -BT 6 (46)	NA	13 (100)	0 (0)	5 (38.6)	14.4
Saeedi et al. (2014) [34]	6	NA	biopsy confirmed recurrence of PCa	Mean 59.5	BT	2 (33)	6 (100)	0 (0)	0 (0)	4.2
Bates et al. (2015) [11]	53	Retrospective	-PSA concentration >0.2 ng/mL -biopsy confirmed recurrence of PCa	67	-EBRT 28 (52.8) -BT 14 (26.4) -IMRT 5 (9.4) -cryotherapy 3 (5.6) -HIFU 3 (5.6)	NA	44 (83)	9 (17)	16 (30.2)	3.7
Pokala et al. (2015) [35]	364	NA	men 40 to 75 years of age with radio-recurrent prostate cancer	Mean 64	-BT -EBRT -or a combination with the both	NA	NA		NA	NA
Pearce et al. (2015) [36]	408	NA	men with adenocarcinoma of the prostate and those who presented with nonmetastatic disease and no nodal involvement	Mean 62.5	-EBRT 348 (89) -BT 43 (11)	NA	167 (63.5)	96 (36.5)	NA	Mean 12.6
Lebdai et al. (2015) [37]	19	Retrospective	biopsy-proven locally persistent or recurrent prostate cancer	64	-Focal VTP	NA	NA		0 (0)	6.3
Mandel et al. (2016) [38]	55	Retrospective	Low comorbidity, life expectancy of at least 10 years, organ-confined PCa <T2b, Gleason score ≤ 7 and preoperative PSA <10 ng/mL	Mean 65.4	-EBRT 27 (49) -HDR 7 (12.7) -LDR 17 (31) -HIFU 4 (7.3)	25 (45)	NA		13 (23.6)	9.5
Vora et al. (2016) [39]	6	Retrospective	-PSA < 10 ng/mL at recurrence -life expectancy > 10 years at recurrence -negative metastatic workup.	64.7	RT	NA	NA		NA	6.08
Kenney et al. (2016) [7]	39	Retrospective	-BCR after radiation therapy -biopsy confirmed Recurrence of PCa	66	-EBRT or PBT 24 (61.5) -BT or BT/EBRT 15 (38.5)	8 (40)	31 (79.5)	6 (15)	14 (36)	Mean 3.5
Orré et al. (2016) [40]	7	NA	Biochemical relapse	66	permanent brachytherapy implants	NA	7 (100)	0 (0)	2 (28.5)	7.13
Vidmar et al. (2017) [41]	24	Retrospective	-recurrent or radiation-resistant prostate cancer	62	-BT 7 (29) -HIFU 7 (29) -EBRT 10 (42)	0 (0)	21 (87)	2 (8)	0 (0)	5.5
Metcalfe et al. (2017) [42]	70	Retrospective	biochemical or biopsy-proven failure	61.06	-EBRT 42 (60) -BT 14 (20) -Proton 6 (8.6) -EBRT + BT 8 (11.4)	18 (26)	60 (88)	8 (11)	10 (14)	5.95
Ogaya-Pinies et al. (2019) [43]	96	Prospective	all patients with a localized, biopsy-proven PCa recurrence after radiotherapy or any ablative technique, with a life expectancy of >10 years	65.75	-EBRT 37 (38.5) -BT 14 (14.5) -EBRT+BT 13 (13.5) -Cyberknife 3 (3) -Proton beam 1 (1) -Cryotherapy 18 (19) -HIFU 7 (8) -Focal VTP 1 (1) -Electroporation 1 (1) -TULSA 1 (1)	NA	NA		NA	4
Onol et al. (2019) [44]	94	Retrospective	-biopsy-proven local recurrence without evidence of metastatic PCa	65	-EBRT 39 (31) -IMRT 15 (12) -PBT 3 (2) -BT23 (18) -combined EBRT + BT 14 (11)	24 (25.5)	91 (97)	3 (3)	36 (38)	Mean 4.53
Devos et al. (2019) [45]	25	Retrospective	-BCR -a positive biopsy following EBRT or BT	65	-EBRT17 (68) -BT 8 (32)	NA	NA		12 (48)	4.6
Mohler et al. (2019) [46]	41	prospective	-biopsy-proven persistent or recurrent CaP diagnosed ≥ 18 months after radiation therapy with PSA ≤ 20 ng/mL -no radiologic evidence of metastatic disease	64	-EBRT 24 (58) -BT 11 (27) -Combined 6 (15)	0 (0)	24 (58.5)	2 (5)	3 (7)	4.1
Clery et al. (2019) [47]	55	NA	-All patients received radiation therapy -BCR was biopsy-proven in all cases	64	-EBRT 30 (55) -BT 10 (18) -HIFU 15 (27)	8 (14.5)	45 (81.5)	2 (4)	3 (5.5)	4.96
Herrera-Caceres et al. (2019) [48]	34	Retrospective	PCa recurrence after focal therapy	61	-Laser ablation 13 (38) -HIFU 19 (56) -Cryotherapy 1 (3) -BT 1 (3)	NA	NA		4 (12)	5.38
Gontero et al. (2019) [49]	395	Retrospective	recurrent PCa	66.3	NA	NA	NA		147 (39)	6.36
De Groote et al. (2020) [50]	106	Retrospective	-All patients received radiation therapy -All patients had biopsy	67	-HIFU 59 (56) -RT27 (25) -BT 10 (9) -ADT 8 (8) -cryotherapy 1 (1) -electroporation /Nanoknife 1 (1)	8 (8)	58 (55)	48 (45)	27 (25)	5.6
Nair et al. (2020) [51]	4	NA	Recurrent CaP	69	transurethral ultrasound ablation (TULSA)	NA	NA		0 (0)	4.3
Thompson et al. (2020) [52]	53	Retrospective	Unsuitable for redo FA (e.g., bilateral/ high-risk cancer) or preference towards radical treatment; Age < 75 yo and fit for major surgery; T1-3aN0M0, surgically resectable on MRI and DRE; Accepting of risks and side effects of surgery.	63	-HIFU	NA	40 (89)	5 (11)	5 (11)	6
Nathan et al. (2021) [53]	135	Retrospective	Primary treatment failure	70	Whole gland therapies: -RT -BT -HIFU Focal gland therapies: -focal HIFU, cryotherapy, and electroporation	NA	80 (55)	61 (45)	35 (26)	5.8
Madi et al. (2021) [12]	26	Retrospective	-All patients had biopsy-proven prostate cancer recurrence.	68.5	-EBRT 18 (69) -BT 4 (15) -Cyberknife 2 (8) -Cryotherapy 2 (8)	NA	NA		13 (50)	5.1
Rajwa et al. (2021) [54]	214	Retrospective	-patients treated with primary radiation therapy-all patients underwent confirmatory biopsy	69	-EBRT 167 (78) -BT39 (18) -EBRT + BT 8 (3.7)	0 (0)	183 (85.5)	30 (14)	86 (40)	3.8
Bozkurt et al. (2021) [55]	10	NA	-clinically organ-confined PCA disease after failure of PBT	66.8	PBT	NA	10 (100)	0 (0)	7 (70)	Mean 5.5
Marra et al. (2021) [56]	414	Retrospective	Recurrent CaP	66	-EBRT 262 (64.5) -BT 106 (25.7) -other primary treatments 56 (13.6)	NA	NA	48 (11.5)	140 (35)	4.2
Spitznagel et al. (2021) [57]	13	Prospective	patients with any detected PCa in the extended follow-up biopsy	61	HIFU	NA	13 (100)	0 (0)	0 (0)	5.4
Wenzel et al. (2021) [58]	428	NA	adult patients (≥18 years old) with histologically confirmed adenocarcinoma of the prostate, diagnosed at biopsy	66	-EBRT: 316 (74) -BT: 67 (16) -EBRT + BT: 45 (10.5)	NA	356 (83)	43 (11)	62 (14.5)	8.8
von Hardenberg et al. (2021) [59]	44	Prospective	biopsy-proven (PCa) after FT	65	-HIFU 42 (95.5) -VTP 2 (4.5)	6 (14)	NA		0 (0)	5.7
Nathan et al. (2022) [60]	100	Prospective Retrospective	locally recurrent prostate cancer after ablative therapy failure	69	-HIFU 92 (92) -Cryotherapy 5 (5) -Electroporation 3 (3)	100 (100)	81 (81)	19 (19)	10 (10)	5.8
Mortensen et al. (2022) [61]	5	Retrospective	-BCR following primary external beam radiation -an expected life expectancy of 10 years or more	71	EBRT	5 (100)	0 (0)	5 (100)	4 (80)	Mean 3.34
Van Riel et al. (2022) [62]	39	Prospective	Recurrent localised PCa	64	irreversible electroporation	NA	39 (100)	0 (0)	0 (0)	6
Blazevski et al. (2022) [63]	15	Retrospective	patients with histopathologically confirmed residual or recurrent clinically signifcant PCa	68	irreversible electroporation	NA	NA		0 (0)	6.6
Catarino et al. (2022) [64]	29	NA	histologically confirmed recurrent PC	65	-BT 9 (31) -EBRT 16 (55) -Cobalt therapy 2 (7) -Tomotherapy 1 (3) -BT+ EBRT 1 (3)	8 (28)	NA		NA	NA

PSA: prostate specific antigen, ADT: androgen deprivation therapy, RT: radiation therapy, sRP: salvage radical prostatectomy, ISUP: International Society of Urological Pathology, NA: not available.

**Table 2 cancers-15-05485-t002:** Intra-operative parameters and pathological features of the overall cohort.

Authors	Surgical Approach *n* (%)	Operative Time (min)	Blood Loss (mL)	Lymph Node Dissection *n* (%)	≥pT3, *n* (%)	sRP ISUP ≥4 *n* (%)	pN+ *n* (%)	PSM *n* (%)	Complications * *n* (%)
Kaouk et al. (2008) [13]	RARP: 4 (100)	125	117	4 (100)	NA	2 (67)	0 (0)	2 (50)	0 (0)
Liatsikos et al. (2008) [14]	LRP: 12 (100)	153	238	8 (66)	4 (33)	5 (42)	0 (0)	3 (25)	1 (8)
Kim et al. (2008) [15]	Open: 5 (71) RARP: 2 (29)	292	914	7 (100)	2 (28.5)	NA	NA	1 (14)	2 (28.5)
Boris et al. (2009) [16]	RARP: 11 (100)	183	113	Standard template 7 (64) Extended template 4 (36)	8 (73)	3 (27)	2 (18)	3 (27)	3 (27)
Seabra et al. (2009) [17]	NA	80	300	NA	11 (26)	6 (14)	NA	NA	Grade 3a: 21 (50) Grade 3b: 2 (4.8)
Leonardo et al. (2009) [18]	Open: 32 (100)	122	550	32 (100)	15 (5)	20 (6)	0 (0)	11 (3)	4 (12.5)
Nunez-Mora et al. (2009) [19]	LRP: 9 (100)	170	250	9 (100)	5 (55.5)	6 (66)	2 (22)	2 (22)	2 (22)
Paparel et al. (2009) [20]	NA	NA	NA	NA	NA	29 (20)	18 (13)	24 (16)	NA
Heidenreich et al. (2010) [21]	Open: 55 (100)	120	360	55 (100)	13 (24)	9 (20)	9 (20)	5 (11)	Grade 1: 13 (23.6) Grade 2: 2 (3.6) Grade 3: 2 (3.6)
Strope et al. (2010) [22]	RARP: 6 (100)	356	280	6 (100)	1 (16)	NA	0 (0)	1 (16)	2 (34)
Eandi et al. (2010) [23]	RARP: 18 (100)	156	150	18 (100)	9 (50)	4 (22)	NA	5 (28)	7 (39)
Chauhan et al. (2011) [24]	RARP: 15 (100)	138	75	12 (80)	10 (77)	7 (47)	1 (6.6)	2 (13)	Grade 1: 1 (7) Grade 2: 1 (7) Grade 3: 1 (7)
Chade et al. (2011) [25]	Open: 404 (100)	NA	NA	58 (14)	NA	96 (24)	65 (16)	99 (25)	NA
Gorin et al. (2011) [26]	Open: 24 (100)	NA	415	15 (63)	13 (54)	NA	2 (13.3)	11 (46)	NA
Ahallal et al. (2011) [27]	Open: 11 (73) RARP: 4 (27)	235	200	15 (100)	9 (60)	7 (47)	2 (13)	2 (13)	Grade 1: 3 (20) Grade 2: 2 (13) Grade 3: 0 (0)
Lawrentschuk et al. (2011) [28]	Open: 15 (100)	135	NA	13 (87)	9 (64)	4 (27)	NA	4 (27)	1 (7)
Leonardo et al. (2012) [29]	LRP: 12 (100)	220	150	13 (100)	8 (61.5)	2 (15)	0 (0)	2 (15)	Grade 1: 3 (23) Grade 2: 1 (8) Grade 3: 2 (15)
Kaffenberger et al. (2013) [30]	RARP: 34 (100)	176	NA	29 (85)	16 (47)	9 (26)	N+: 0 (0) Nx: 5 (15)	9 (26)	Grade 1: 11 (32) Grade 2: 1 (3) Grade 3: 1 (3)
Peters et al. (2013) [31]	Open: 44 (100)	NA	NA	NA	NA	NA	NA	NA	NA
Yuh et al. (2014) [32]	RARP: 51 (100)	179	175	43 (84)	26 (51)	11 (21.6)	3 (6)	16 (31.4)	Grade 1 2: 13 (25.5) Grade 3 4: 22 (43)
Zugor et al. (2014) [33]	RARP: 13 (100)	154	130	13 (100)	6 (46)	7 (54)	0 (0)	0 (0)	Minor complications 4 (30.7) Grade 1: 2 (15.3) Grade 3a: 2 (15.3) Major complications 0 (0)
Saeedi et al. (2014) [34]	Open: 6 (100)	NA	NA	6 (100)	1 (17)	0 (0)	1 (17)	2 (33)	2 (33)
Bates et al. (2015) [11]	RARP: 53 (100)	128 Console time: 80	100	NA	26 (49)	19 (36)	NA	10 (19)	Grade 1 2: 1 (2) Grade 3 4: 0 (0)
Pokala et al. (2015) [35]	NA	NA	NA	286 (79)	186 (51)	NA	40 (11)	NA	NA
Pearce et al. (2015) [36]	NA	NA	NA	273 (75)	169 (49)	19 (6.2)	122 (30)	124 (34)	NA
Lebdai et al. (2015) [37]	Open: 12 (63) RARP: 5 (26) LRP: 2 (11)	150	400	19 (100)	7 (37)	1 (5)	1 (5)	9 (47)	Grade 1: 1 (5) Grade 2: 1 (5) Grade 3: 1 (5)
Mandel et al. (2016) [38]	Open: 55 (100)	NA	725	55 (100)	22 (40.5)	13 (23.6)	12 (22)	15 (27.5)	Grade 3: 7 (12.7)
Vora et al. (2016) [39]	RARP: 6 (100)	NA	NA	NA	NA	NA	NA	NA	1 (16.7)
Kenney et al. (2016) [7]	Open: 19 (49) RARP: 20 (51)	297	623	39 (100)	24 (61.5)	18 (46)	5 (13)	6 (15)	Grade 1 2: 43 (77) Grade 3 4: 13 (23)
Orré et al. (2016) [40]	RARP: 7 (100)	142	NA	2 (28.5)	5 (71)	NA	4 (57)	1 (14)	2 (28.5)
Vidmar et al. (2017) [41]	RARP: 12 (50) Open: 12 (50)	180	300	4 (33)	7 (63)	4 (44)	1 (8)	6 (50)	0 (0)
Metcalfe et al. (2017) [42]	NA	NA	NA	70 (100)	42 (60)	14 (20)	38 (54)	14 (20)	NA
Ogaya-Pinies et al. (2019) [43]	RARP: 96 (100)	125	100	85 (89)	22 (23)	8 (8)	29 (30)	16 (17)	Grade 1: 20 (21) Grade 2: 1 (1) Grade 3: 3 (3) Grade 4: 1 (1)
Onol et al. (2019) [44]	RARP: 126 (100)	129 Console time: 84	107	94 (100)	47 (50)	40 (42.6)	10 (10.6)	16 (17)	Clavien 1: 9 (9.7) Clavien 2: 11 (11.8) Clavien 3a: 2 (2.2) Clavien 3b: 1 (1.1) Clavien 4a: 1 (1.1)
Devos et al. (2019) [45]	Open: 23 (92) RARP: 2 (8)	166	808	23 (92)	14 (56)	12 (48)	7 (28)	11 (44)	22 (100) Grade 1: 1 (4) Grade 2: 5 (20) Grade 3: 16 (64)
Mohler et al. (2019) [46]	Open: 41 (100)	213	NA	41 (100)	23 (57)	24 (58.5)	5 (12)	7 (17)	44 (100)
Clery et al. (2019) [47]	RARP: 44 (80) Open: 11 (20)	150	300	55 (100)	31 (56)	15 (27)	6 (11)	4 (7)	Grade 1: 44 (80) Grade 2: 1 (1.8) Grade 3: 2 (2.3)
Herrera-Caceres et al. (2019) [48]	Open: 28 (82) LRP: 1 (3) RARP: 5 (15)	NA	512	34 (100)	20 (59)	2 (6)	NA	13 (38)	Intraoperative complications: Cystotomy 2 (6)
Gontero et al. (2019) [49]	Open: 186 (47) RARP: 209 (53)	221	468.5	337 (85)	217 (55)	170 (43)	63 (16)	NA	146 (37)
De Groote et al. (2020) [50]	RARP: 106 (100)	142	200	NA	70 (66)	23 (22)	NA	RT: 14 (52)	8 (8) Grade 3a: 1 (1)
Nair et al. (2020) [51]	Open: 4 (100)	210	866	4 (100)	3 (75)	1 (25)	NA	2 (50)	1 (25)
Thompson et al. (2020) [52]	RARP: 53 (100)	Console time: 140	200	NA	34 (64.5)	5 (11)	NA	23 (44)	Grade 1: 4 (9) Grade 2: 3 (7) Grade 3: 1 (2)
Nathan et al. (2021) [53]	RARP: 135 (100)	165	200	25 (18.5)	77 (57)	26 (29)	NA	51 (38)	Grade 1: 9 (7) Grade 2: 7 (5) Grade 3–5: 2 (1.5)
Madi et al. (2021) [12]	RARP: 26 (100)	170.5	75	26 (100)	11 (42)	15 (58)	1 (4)	8 (31)	4 (15) Grade 1: 1 (4) Grade 2: 0 (0) Grade 3: 3 (11)
Rajwa et al. (2021) [54]	NA	198	600	214 (100)	159 (74)	86 (40.1)	40 (19)	43 (20)	Grade 1: 21 (9.8) Grade 2: 167 (78) Grade 3: 26 (12)
Bozkurt et al. (2021) [55]	RARP: 10 (100)	230.7	745	10 (100)	8 (80)	6 (60)	2 (20)	2 (20)	Grade 1–2: 19 (90) Grade 3–4: 5 (30)
Marra et al. (2021) [56]	Open: 216 (52) RARP: 198 (48)	186.5	300	349 (84.3)	218 (53)	151 (40)	65 (16.0)	122 (29.7)	Grade 1–2: 144 (41.5) Grade 3–4: 65 (19)
Spitznagel et al. (2021) [57]	RARP: 13 (100)	260	230	13 (100)	3 (23)	1 (8)	1 (8)	1 (8)	Grade 1: 2 (15) Grade 2: 0 (0) Grade 3: 4 (31)
Wenzel et al. (2021) [58]	NA	NA	NA	428 (100)	47 (11)	17 (4)	24 (6)	NA	NA
von Hardenberg et al. (2021) [59]	Open: 16 (36) LRP: 3 (7) RARP: 25 (57)	NA	NA	NA	14 (32)	16 (36)	3 (7)	10 (23)	NA
Nathan et al. (2022) [60]	RARP: 100 (100)	170	200	NA	NA	NA	NA	38 (38)	Grade 1: 6 (6) Grade 2: 2 (2) Grade 3: 1 (1)
Mortensen et al. (2022) [61]	RARP: 5 (100)	205	120	0 (0)	3 (60)	NA	3 (60)	3 (60)	Grade 1: 3 (60) Grade 2: 1 (20)
Van Riel et al. (2022) [62]	LRP: 3 (8) RARP: 36 (92)	NA	182	9 (23)	18 (46)	8 (21)	0 (0)	10 (26)	NA
Blazevski et al. (2022) [63]	RARP: 15 (100)	NA	200	4 (27)	6 (40)	6 (40)	1 (7)	1 (7)	0 (0)
Catarino et al. (2022) [64]	LRP: 29 (100)	90	200	25 (86)	19 (65.5)	13 (45)	5 (17)	8 (28)	Grade 2: 4 (14) Grade 3: 3 (10)

NA: not available, sRP: salvage radical prostatectomy, PSM: positive surgical margins, RT: radiation therapy. * According to classification of Clavien-Dindo.

**Table 3 cancers-15-05485-t003:** Oncological and functional outcomes of the overall cohort.

Authors	Median Follow-Up (Months)	BCR *n* (%)	Recurrence Free Survival (RFS) (%)	Cancer-Specific Survival (%)	Overall Survival (%)	Urinary Continence *n* (%)
Kaouk et al. (2008) [13]	5	NA	NA	NA	NA	At 1 month: 3 (75)
Liatsikos et al. (2008) [14]	Mean 20	At 12 months: 1 (8)	NA	NA	NA	10 (83)
Kim et al. (2008) [15]	NA	NA	NA	NA	NA	NA
Boris et al. (2009) [16]	20.5	At 43 months: 3 (30)	NA	NA	NA	6 (54.5)
Seabra et al. (2009) [17]	18	NA	NA	NA	NA	12 (28)
Leonardo et al. (2009) [18]	35	8 (25)	NA	NA	NA	At 1 year: 0 pads per day: 7 (22) 1–2 pads per day: 20 (62.5)>2 pads per day: 5 (15.5)
Nunez-Mora et al. (2009) [19]	26.8	At 16 months: 2 (22)	NA	NA	NA	complete continence:3 (33) 1–2 pads per day:4 (44)
Paparel et al. (2009) [20]	4.6 y	65 (44.5)	5 year: 54	NA	NA	NA
Heidenreich et al. (2010) [21]	23	NA	NA	NA	NA	At 1 year: complete continence: 44 (80)
Strope et al. (2010) [22]	15	At 6 weeks: 2 (34)	NA	NA	NA	At 1 year: 2.3 pads per day: 4 (66)
Eandi et al. (2010) [23]	18	At 18 months: 2 (33)	NA	NA	NA	6 (33)
Chauhan et al. (2011) [24]	4.6	At 5 months: 4 (28.6)	NA	NA	NA	11 (71.4)
Chade et al. (2011) [25]	4.4 y	At 5y: 48	NA	At 10y: 83	NA	NA
Gorin et al. (2011) [26]	63	At 2y: 14 (58)	5 year: 40	NA	5 year: 90	23 (96)
Ahallal et al. (2011) [27]	8	NA	NA	NA	NA	0 pads per day: 7 (47) 1–2 pads per day: 7 (47)
Lawrentschuk et al. (2011) [28]	NA	NA	NA	NA	NA	At 1 year:0 pads per day: 6 (60)
Leonardo et al. (2012) [29]	14	at 10 months: 1 (8)	NA	NA	NA	0 pads per day: 9 (69)2 pads per day:4 (31)
Kaffenberger et al. (2013) [30]	16.1	At 16 months: 6 (18)	NA	NA	NA	20 (65)
Peters et al. (2013) [31]	60	At 22 months: 29 (66 )	NA	NA	NA	NA
Yuh et al. (2014) [32]	36	At 3 years: 57	NA	NA	5 year: 100	At 6 month: 23 (45)
Zugor et al. (2014) [33]	23	3 (23)	NA	NA	NA	At 12 month: 7 (54)
Saeedi et al. (2014) [34]	NA	NA	NA	NA	NA	At 12 month: 0 pads per day: 5 (83) 1 pads per day:1 (17)
Bates et al. (2015) [11]	26	At 13 months: 8 (15)	NA	NA	NA	At 36 month: 41 (77)
Pokala et al. (2015) [35]	NA	NA	NA	10 years: 88.6	10 years: 77.5	NA
Pearce et al. (2015) [36]	NA	NA	NA	NA	NA	NA
Lebdai et al. (2015) [37]	NA	NA	NA	NA	NA	At 10 month: Completely continent: 13 (68) ≤1 pad/day: 5 (27) 3 pads/day: 1 (5)
Mandel et al. (2016) [38]	36	23 (42)	5-year: 48.7	NA	5-year: 88.7	41 (74)
Vora et al. (2016) [39]	NA	NA	NA	NA	NA	1 (16.7)
Kenney et al. (2016) [7]	16.8	NA	9.5 months robotic	NA	NA	4 (10)
Orré et al. (2016) [40]	NA	NA	NA	NA	NA	At 12 month: 4 (57)
Vidmar et al. (2017) [41]	25	NA	NA	NA	NA	7 (30)
Metcalfe et al. (2017) [42]	2.79 y	At 5 months: 35 (51.5)	Median: 2.78	NA	NA	NA
Ogaya-Pinies et al. (2019) [43]	14	At 1 year: 15 (16)	NA	NA	NA	At 12 month: 0 pads per day: 55 (57) 1–2 pads per day: 25 (26)
Onol et al. (2019) [44]	32	16 (17) Radiation group 6 (19) Focal ablation group	5-year: 56	NA	NA	At 1 year Overall full (no pads): 37 (39.2) social (0–1 pad/day): 48 (51.3)
Devos et al. (2019) [45]	43	NA	NA	5-year: 74	5-year: 62	4 (16)
Mohler et al. (2019) [46]	91	NA	At 10y: 33	NA	At 10y: 52	At 12 year: 6 (15)
Clery et al. (2019) [47]	24	At 13 months: 17 (31)	NA	5 years: 80	NA	27 (49.1)
Herrera-Caceres et al. (2019) [48]	52	At 42 months: 7 (21)	NA	NA	NA	≤1 pad: 31 (91) ≥2 pads: 2 (6)
Gontero et al. (2019) [49]	3 years	NA	NA	NA	NA	At 12 month: fully continent: 221 (56)
De Groote et al. (2020) [50]	2.1 years	At 25 months: 26 (24)	5-year: 60	NA	NA	At 2 years or more fully continent: 53 (50) socially continent: 35 (33)
Nair et al. (2020) [51]	43	2 (50)	NA	NA	NA	Continent:1 (25) 0–1 pads per day: 2 (50)
Thompson et al. (2020) [52]	18	At 3 months: 8 (16)	NA	NA	NA	Pad-free at 12-months: 35 (65.5) Socially continent at 12-mo (0–1 pad): 46 (86)
Nathan et al. (2021) [53]	17	At 26 months: 17 (33)	5 years: 60	NA	129 (96)	At 12 month: fully continent: 90 (67)
Madi et al. (2021) [12]	18	NA	NA	NA	NA	14 (100)
Rajwa et al. (2021) [54]	25.3	NA	NA	NA	NA	NA
Bozkurt et al. (2021) [55]	32	NA	NA	NA	NA	0–1 pads per day: 2 (20) 2 pads per day: 6 (60)
Marra et al. (2021) [56]	36	At 12 months: 115 (31)	NA	5 years: 98	5 years: 92	85 (28.2)
Spitznagel et al. (2021) [57]	12	At 12 months: 1 (8)	NA	NA	NA	No incontinence: 3 (20) Mild incontinence: 3 (20) Moderate incontinence: 6 (50)
Wenzel et al. (2021) [58]	74	NA	NA	5 years: 13.4	NA	NA
von Hardenberg et al. (2021) [59]	28	NA	3 years: 80	NA	NA	NA
Nathan et al. (2022) [60]	16.5	At 16.5 months: 31 (23)	5 years: 75	NA	NA	At 12 month: 77 (77)
Mortensen et al. (2022) [61]	13	NA	NA	NA	NA	5 (100)
Van Riel et al. (2022) [62]	17.7	At 6 months: 1 (2.5)	NA	100	100	34 (94.4)
Blazevski et al. (2022) [63]	22	At 22 months: 0 (0)	NA	NA	NA	Pad free at 3 months: 14 (93) Pad free at 6 months: 1 (7)
Catarino et al. (2022) [64]	94	At 61 months: 17 (59)	5 years: 50	NA	NA	At 12 month: Pad-free continence: 6 (21) Mild incontinence: 12 (41)

NA: not available, RFS: recurrence-free survival, BCR: biochemical recurrence.

## Data Availability

The data are contained within the article.

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
