# Peer review of "Salvage Radical Prostatectomy for Recurrent Prostate Cancer: A Systematic Review (French ccAFU)"

_cancers, 2023, doi:10.3390/cancers15225485_

Round 1

Reviewer 1 Report

Comments and Suggestions for Authors

It was a pleasure reviewing the manuscript "Salvage Radical Prostatectomy for Recurrent Prostate Cancer: A Systematic Review (French ccAFU)"

The authors present a systemic review of 55 studies including 3,836 patients. which met the eligibility criteria. The vast majority of men included had Radiation therapy (including Brachytherapy) as first-line treatment (n=3,240, 84%). Other first-line treatments included HIFU (n=338, 9%), electroporation (n=59, 2%), Proton Beam Therapy (n=54, 1.5%), cryotherapy (n=34, 1%), focal Vascular Targeted Photodynamic therapy (n=22, 0.6%), and transurethral ultrasound ablation (n=19, 0.5%). Median preoperative PSA at the time of recurrence ranged from 1.5 to 14.4 ng/mL. The surgical approach was open in 2300 (60%), robotic in 1465 (38%) and laparoscopic in 71 (2%) cases. Since 2019, there was an increase of robotic versus conventional surgery (1245 versus 525 of cases, respectively). Concomitant lymph node dissection was performed in 2587 cases (79%). The overall complication rate was 34%. Biochemical recurrence ranged from 8% to 51.5% at 12 months, from 0% to 66% at 22 months, from 48% to 59% at 60 months. Specific and overall survival rates ranged from 13.4 to 98% and 62 to 100% at 5 years, respectively. Urinary continence was maintained in 52.1% of cases. 

I have following suggestion:-

1. There have been similar systematic reviews which have already been published. The authors should acknowledge them and describe how their review is different.

Author Response

We would like to thank the reviewer for taking the time to decorticate our review in detail

Our systematic review is considered to be the most exhaustive, including 55 studies as well as all non-surgical therapies prior to recurrence.

Reviewer 2 Report

Comments and Suggestions for Authors

Congratulations to the authors for the effort in reviewing the literature about this subject which lack of strong evidence. The authors have performed a well-designed and previously registered systematic review.

Only minor comments should be addressed:

Some information in figure 1 is not visible.

In oncological results the last paragraph about BCR rate and survival indirect information of each approach should include the follow up or specific moment where they are evaluated as the open approach cohorts had longer follow up

Author Response

I thank the reviewer very much for the quality of his reviewing.

We have corrected the figure 1

We have added the follow up of each surgical approach

The median follow-up ranged from 4.6 to 94 months (36, 22 and 39.5 months respectively in laparoscopic, robotic and open approaches).

Reviewer 3 Report

Comments and Suggestions for Authors

The Manuscript is a systemic review about salvage radical prostatectomy for recurrent prostate cancer. It is well written and interesting but some minor points should be highlighted:

1) A risk of bias table should be added;

2) the patient selection flowchart should be modified in order to show all the text (especially the "exclusion criteria: abstract screening" box).

Author Response

I thank the reviewer very much for the quality of his reviewing.

We have added this risk of bias table and we have modified our flowchart.

Reviewer 4 Report

Comments and Suggestions for Authors

Salvage procedures following non-surgical procedures for prostate cancer are subject to many subjective issues. The studies described are all retrospective and are all a selection of patients and differ in non-surgical approach, which makes interpretation difficult. You can imagine that patients undergoing focal therapy have different characteristics compare to whole gland approaches and thus outcome of salvage RP could be different.

Furthermore, the time span of included publications is quite long, which could influence results.

What are also important factors is time to recurrence following initial treatment and clinical workup of patients at time of recurrence that should be addressed

Author Response

We thank the reviewer very much for the quality of these pertinent remarks. We completely agree with you as mentioned in the limitations section.

Our study has several strengths, including the important number of studies/patients included, with a variety of nonsurgical primary treatments with a clear distinction between them, the inclusion of most updated data and the careful review for study inclusion.